# Room-temperature spontaneous superradiance from single diamond nanocrystals

Carlo Bradac[1,2], Mattias T. Johnsson[1,2], Matthew van Breugel[1,2], Ben Q. Baragiola[1,2], Rochelle Martin[1,2], Mathieu L. Juan [1,2], Gavin K. Brennen[1,2] & Thomas Volz[1,2]

Superradiance (SR) is a cooperative phenomenon which occurs when an ensemble of quantum emitters couples collectively to a mode of the electromagnetic field as a single, massive dipole that radiates photons at an enhanced rate. Previous studies on solid-state systems either reported SR from sizeable crystals with at least one spatial dimension much larger than the wavelength of the light and/or only close to liquid-helium temperatures. Here, we report the observation of room-temperature superradiance from single, highly luminescent diamond nanocrystals with spatial dimensions much smaller than the wavelength of light, and each containing a large number ($\sim 10^3$) of embedded nitrogen-vacancy (NV) centres. The results pave the way towards a systematic study of SR in a well-controlled, solid-state quantum system at room temperature.

[1] Department of Physics & Astronomy, Macquarie University, NSW 2109, Australia. [2] ARC Centre of Excellence for Engineered Quantum Systems, Macquarie University, NSW 2109, Australia. Carlo Bradac, Mattias T. Johnsson and Matthew van Breugel contributed equally to this work. Correspondence and requests for materials should be addressed to T.V. (email: thomas.volz@mq.edu.au)

The occurrence of SR in an ensemble of identical emitters indicates the build-up of large-scale coherence between individual dipoles. Due to the many available pathways for photon emission from a system of $N$ indistinguishable initially excited emitters, the de-excitation process itself leads to the formation of highly entangled symmetric superposition states, so-called Dicke states[1]. Dicke described the system of $N$ dipoles (or two-level emitters) using collective pseudospin operators with total spin $J = N/2$ and projection $M$, corresponding to $J + M$ excitations. The fluorescence rate from state $|J, M\rangle$ is $\gamma_{J,M} = \gamma(J(J+1) - M(M-1))$ with $\gamma$ being the single-dipole emission rate. As the system cascades down the Dicke ladder of states, the photon emission rate scales at maximum as $N^2$, an enhancement by a factor $N$ over independent dipoles, hence the name superradiance.

As set forth by Dicke in his seminal 1954 paper[1], the conditions required for SR arise from the indistinguishability of the quantum emitters with respect to the field mode. Spatial indistinguishability occurs when the emitters are confined to a volume much smaller than the scale set by the wavelength of the emitters' optical transition, $V \ll \lambda^3$. In addition, the emitters must be spectrally indistinguishable, which complicates the study of SR in typical solid-state settings due to large inhomogeneous broadenings and unavoidable dephasing.

Observed initially in well-isolated, controlled laboratory settings[2, 3], SR has over time found applications in a variety of fields. For instance, it has been evoked as an underlying mechanism for exciton delocalisation in light-harvesting complexes[4]. In astrophysics, SR is predicted to occur in the vicinity of black holes[5], and in the field of precision metrology, a novel superradiant laser source was realised promising unprecedented narrow linewidths[6]. Further, the presence of highly entangled multi-particle states is an attractive prospect for quantum metrology[7, 8]. Dicke states possess symmetries that render them immune to certain types of environmental noise that affect all emitters in the same way and cannot resolve individual dipoles[9]. In a solid-state setting, this includes global dephasing due to long-wavelength phonon modes. However, local dephasing mechanisms, such as coupling to short-wavelength phonons and coupling to electric fields arising from ionisation or other local defects, can have a detrimental effect on the cooperative behaviour of a system. Indeed, it is the simultaneous requirement of high emitter density and low local decoherence that has made SR challenging to observe at room temperature in solid-state or atomic systems.

Here, we report the observation of room-temperature superradiance from single, highly luminescent nanodiamonds (NDs) with spatial dimensions much smaller than the wavelength of light, and each containing a large number ($\sim 10^3$) of embedded NV centres. Under pulsed off-resonant excitation, we observe (i) ultrafast radiative lifetimes down to around 1 ns, and (ii) super-Poissonian photon statistics of the light emitted from the fastest NDs. We explain our observations with a detailed theoretical

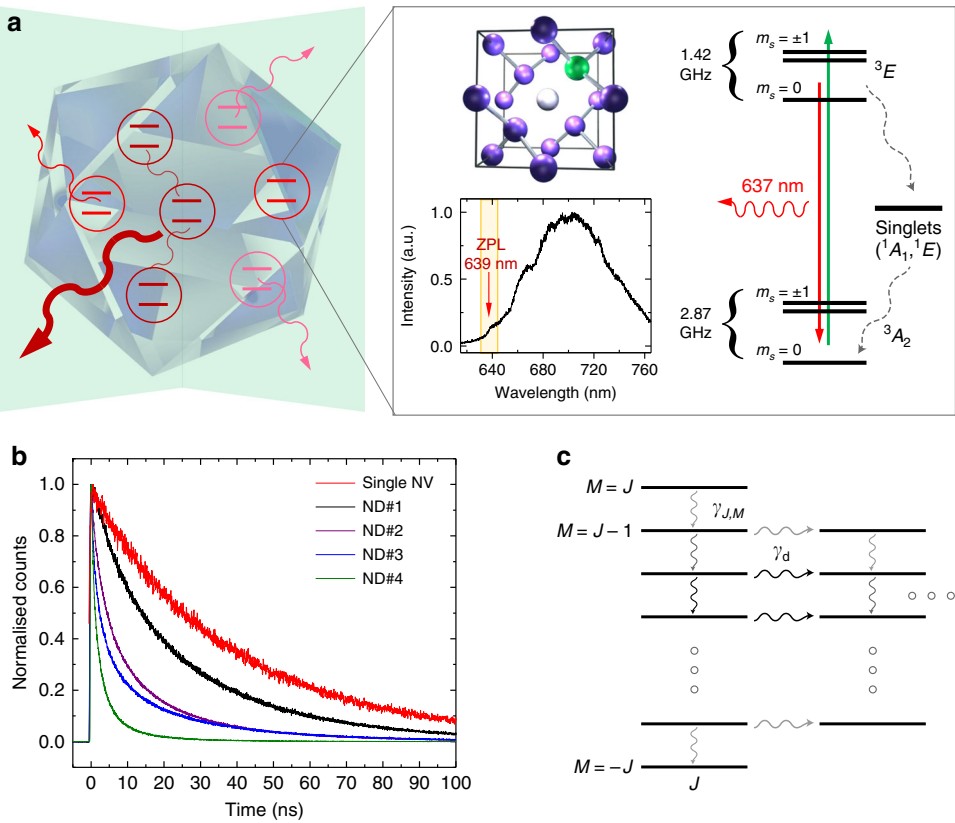

**Fig. 1** Superradiance from nanodiamonds with many NV centres. **a** Graphic representation of cooperatively interacting NV centres emitting a superradiant burst from a nanoscale diamond crystal. The zoom-in shows the underlying crystalline structure around a single NV centre, with the substitutional nitrogen atom indicated in green and the vacancy in white. It also displays the level structure of a single NV centre in bulk diamond[11]. Due to strong vibronic sidebands, only a fraction of photons is emitted into the ZPL. In our high-density NV ND sample the ZPL is shifted from the bulk value of 637 nm, as seen in the representative fluorescence spectrum with a ZPL of around 639 nm[12]. **b** Measured normalised fluorescence decay curves for five different NDs, with lifetimes ranging from the usual few tens of nanoseconds for a single NV centre in a ND (red trace), to around 1 ns for high-density NV NDs (green trace, ND #4). **c** Illustration of the Dicke ladder of states: Collective optical decay couples descending states within each pseudospin $J$-subspace at a characteristic rate $\gamma_{J,M}$. Local dephasing at rate $\gamma_d$ decouples individual spins from the collective subspace, leaving the remaining spins in a smaller $J$-subspace. Thicker/darker decay lines denote stronger decay rates with maximum decay near $M = 0$ states

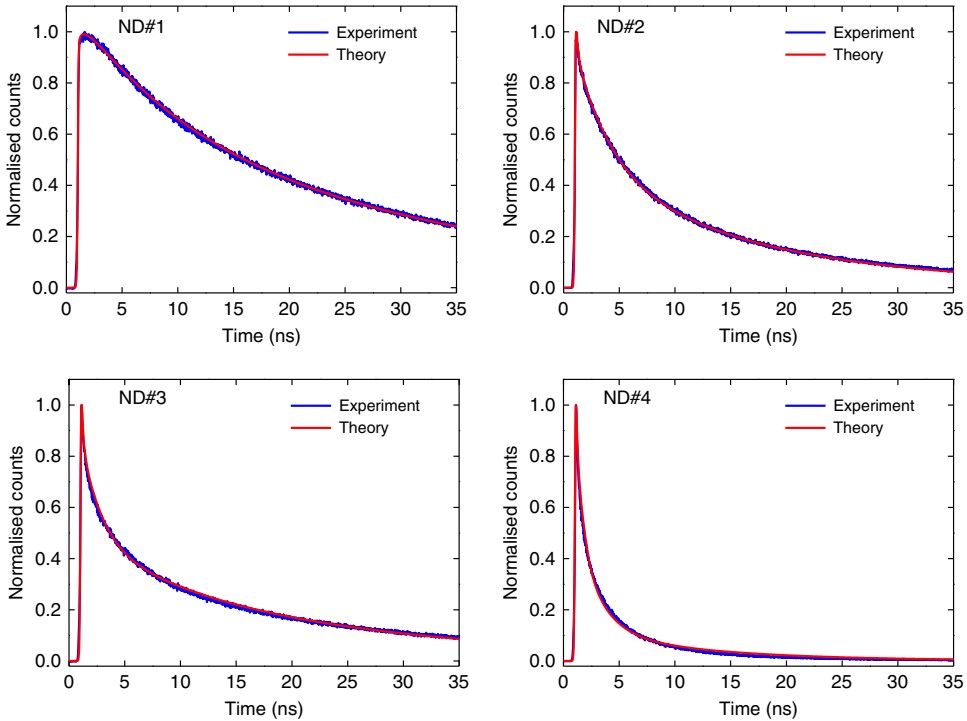

**Fig. 2** Fluorescence decay curves and corresponding fits for four different NDs. The four graphs display measured fluorescence decay curves (blue) with corresponding fits (red) obtained from our model (see main text), showing excellent agreement. Note that the curves are normalised to their respective maximum. The different NDs exhibit increasingly faster photo-emission, with corresponding lifetimes of {25, 3.6, 2.2, 1.1} ns for ND#1–4. The shorter lifetimes correspond to larger collective domain sizes of $N_{max} = \{2, 7, 10, 50\}$, respectively (Supplementary Note 2)

model that suggests cooperative domains of up to 50 NV centres for the fastest NDs. Our results open a promising route towards a systematic study of SR in a solid-state quantum system at room temperature. Ultimately, quantum engineering of superradiance in diamond could have applications in quantum sensing, energy harvesting and efficient photon detection[10].

## Results

**Radiative lifetime measurements**. The NV centre (Fig. 1a) in diamond is an extrinsic defect where two adjacent carbon atoms in the lattice are replaced by a substitutional nitrogen atom and a vacancy[11]. Its most stable form, the negatively charged $NV^-$, displays triplet electronic ground ($^3A_2$) and optically excited ($^3E$) states, and intermediate singlet states ($^1A_2$ and $^1E$). The separation in energy (1.945 eV) between the ground and the excited states ($^3A$–$^3A$) corresponds to a zero phonon line (ZPL) at 637 nm, followed by characteristic phononic sidebands associated with local vibrational modes. These local vibrational modes are due to deformations in the lattice within a few unit cells of the defect and are characterised by ultrafast femto/picosecond, non-radiative relaxation. Previous work[13] suggested that these local modes decay into global, long-wavelength, acoustic phonon modes which exhibit decay on a much longer timescale of a few tens of picoseconds, which in turn is still much shorter than any optical rate in the system. The decay of local into global phonons erases any information that the local environment would have gained and therefore ultimately preserves the coherence among the emitters and enables the subsequent superradiant photon emission (Supplementary Note 1).

From an experimental point of view, the most salient feature of SR is accelerated optical emission, whose intensity burst can scale faster than linearly with the number of emitters. To investigate this phenomenon, we measured fluorescence decay of 100 separate NDs hosting a high density of NV centres ($\sim 3 \times 10^6$ NV

centres per $\mu m^3$, see Methods) and compared the decay curves against our theoretical model (see below). In addition, we measured brightness and size for the 100 NDs, and performed saturation-intensity measurements. Figure 1b shows a subset of NV decay curves representative of NDs of different size and brightness. The red curve shows the exponential decay for a single NV centre used for reference. The faster diamonds were not well fit by exponentials, but rather required a superradiant model, as described below. However, to make speed comparisons we extracted indicative $1/e$-lifetimes by fitting a standard exponential to the first three nanoseconds of each decay curve. For the fastest diamonds we observed lifetimes around 1 ns or even below, never reported before for NV centres (e.g., Figure 1b, ND#4).

**Determining the number of cooperatively scattering NVs**. We developed a detailed theoretical model to describe the SR (Supplementary Note 1). Briefly, we assume a collection of individual spectral domains (most likely corresponding to spatial domains within the nanocrystal), each containing a different number of NV centres that initially act collectively. The $m_s = 0$ and $m_s = \pm 1$ populations are treated as two separate collections of domains. We further include the inter-system crossing (ISC), which mixes the spin-state populations, and due to its non-radiative nature decreases the overall collective radiation with a rate $\gamma_{ISC}$. The collective behaviour breaks down over time due to local dephasing at rate $\gamma_d$, projecting the collective centres partially into a lower-dimensional collective subspace and partially into the non-collective space that undergoes standard exponential decay.

The inputs to the model are: (i) the number $N$ of NV centres in each collective domain, (ii) the initial state of each domain, (iii) the underlying bright (radiative) and dark (non-radiative) decay rates, as well as (iv) the local dephasing rates for the $m_s = 0$ and $m_s = \pm 1$ populations. The dark decay rates from the ISC are not well known; we take the best estimates from[11] and use

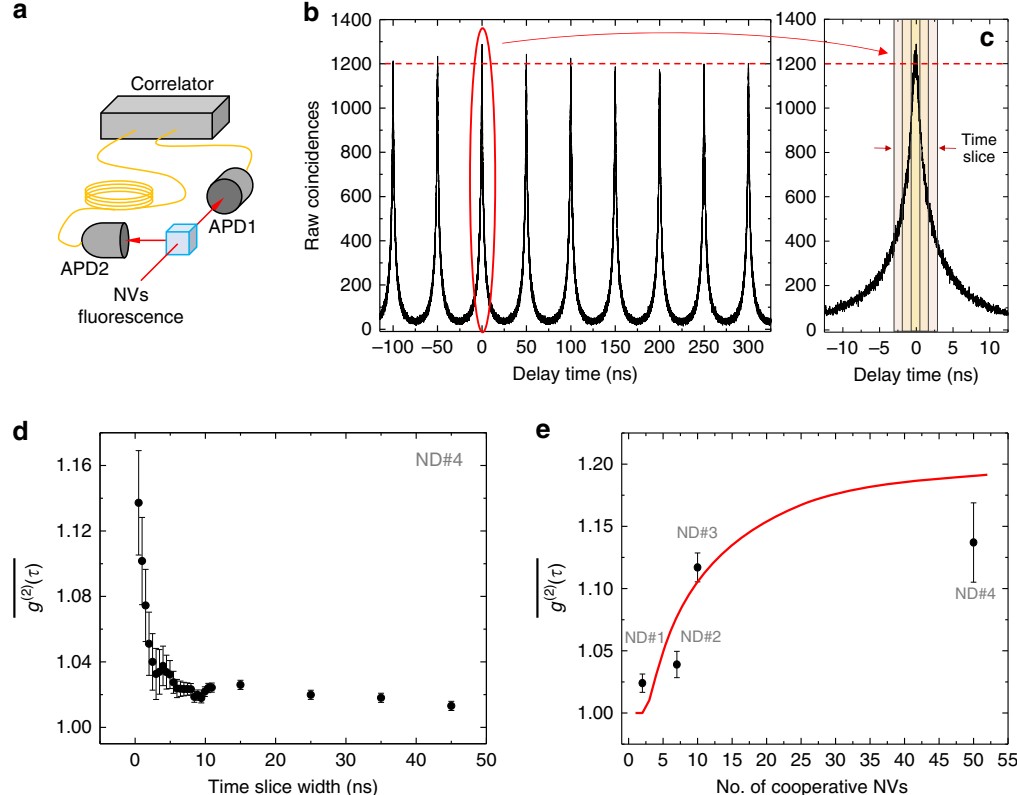

**Fig. 3** Autocorrelation measurements. **a** Schematic representation of the Hanbury–Brown and Twiss interferometer. **b** Normalised coincidences for ND#4. **c** Time slicing employed to evaluate the autocorrelation function. **d** Measured time-integrated autocorrelation function $\overline{g^{(2)}(\tau)}$, which approaches $g^{(2)}(0)$ as $\tau \rightarrow 0$. For ND#4, $\overline{g^{(2)}(0)}$ crests at 1.14 ± 0.02 for a time-slice width of 0.5 ns; it drops considerably as the width increases above 2–3 ns (after which the SR burst has exhausted) to then converge to Poissonian/random photoemission $\overline{g^{(2)}(\tau)} \sim 1$ expected from many NV centres at long times. Error bars are determined from the standard error of the area under the peaks, for each set of time slices, excluding the '0' peak. **e** Measured maximum value of $\overline{g^{(2)}(0)}$ for ND#1–4, and corresponding theoretically estimated number $N_{max}$ of NV centres acting cooperatively to produce such value of $\overline{g^{(2)}(0)}$ using the initial state assumed by our model. Because of the finite time window for averaging (~ 0.5 ns) this is an underestimate of $g^{(2)}(0)$. The continuous line (red) sets the upper limit of our theoretical prediction for $g^{(2)}(0)$

$\gamma_{\mathrm{ISC}}^0/2\pi = 1.8\,\mathrm{MHz}$ and $\gamma_{\mathrm{ISC}}^{\pm 1}/2\pi = 9.4\,\mathrm{MHz}$ for the $m_s = 0$ and $m_s = \pm 1$ rates, respectively. The bright decay rates are heavily influenced by the size and geometry of each individual ND, meaning we cannot simply use bulk rates. To obtain the bright rate for each ND we perform an exponential fit on the long-term tail of the decay curve, well after the collective processes have ended. This leaves the local dephasing rates, the number of centres in each domain, and the initial state of the collective space as free parameters in the model. To further constrain the model, we chose a particular probability distribution (Supplementary Note 2) for the number of NV centres across the collective domains. For fitting purposes, this distribution was characterised by a single free fit parameter $N_{max}$ corresponding to the maximum domain size. In addition, we assume that the initial state consists of having each $M$-level in the symmetric Dicke ladder equally populated (Fig. 1c). This initial state assumption is not critical, as different distributions across the $M$-levels can provide equally good fits by making small changes to $N_{max}$.

With these assumptions, we find excellent agreement between the fits from our model and the fluorescence decay curves of each of the 100 NDs we characterised (see Supplementary Note 2). Figure 2 shows the fits for four NDs (ND#1–4) representative of four distinct typical decay rates, each corresponding to a different collective domain size with faster decay indicating larger domain size. The local dephasing rates extracted from the fits were largely consistent across all the NDs and varied between $\gamma_{\mathrm{d}}^0/2\pi \sim 20 - 40\,\mathrm{MHz}$ and $\gamma_{\mathrm{d}}^{\pm 1}/2\pi \sim 300 - 450\,\mathrm{MHz}$ for the

$m_s = 0$ and $\pm 1$ domains, respectively. We attribute the roughly ten times higher dephasing rates for the $m_s = \pm 1$ states to inhomogeneous electric fields throughout the crystal (Supplementary Note 3). The fits also allow for the extraction of the initial NV spin polarisation, i.e., the fraction of spins initially in the $m_s = 0$ state. The measured spin polarisation of ~ 50–60% across the investigated NDs is in line with previous measurements of spin polarisation in high-density NV samples[14, 15].

For the majority of the NDs, we found a typical cooperative domain size of $N_{max} \sim 1$–2, indicating absent or very little collective behaviour. However, the faster decaying diamonds (e.g., ND#2–4) were accurately fitted by using a higher mean number of centres acting collectively ($N_{max} \sim 10$–50), as shown in Fig. 2c, d. We note that collective superradiance requires indistinguishability of the emitters, which can be broken in many ways in a solid-state setting. The fact that the majority of NDs in our ensemble did not exhibit superradiant behaviour suggests that environment seen by the NV centres included many inhomogeneous effects. We attribute the majority of the inhomogeneity to the material preparation which involves high-dose proton irradiation followed by annealing. However, stochastic variation gives rise to the existence of a few NDs exhibiting domains with large numbers of spatially and spectrally identical NV centres which do act cooperatively.

It should be noted that previous studies reported a decrease in the lifetimes of NVs for centres produced via low-energy He-ion irradiation, with the decay time decreasing for increasing ion

doses. This effect has been attributed to increased damage in the crystal lattice which provides nonradiative decay paths with faster dynamics[16, 17]. This is however inconsistent with our observations where we found that higher peak fluorescence correlated to faster decay rates (Supplementary Note 5)—the exact opposite of what would be expected if the shortening of the lifetimes was indeed due to non-radiative, dark pathways. To test quantitatively against other possible explanations for the observed fast decay dynamics, we also attempted to fit the observed fluorescence decay curves with both a bi-exponential and a deformed exponential[18] decay, both of which gave clearly worse results.

**Photon autocorrelation measurements**. To collect further experimental evidence for the validity of our theoretical model, we performed autocorrelation measurements by means of a Hanbury–Brown and Twiss interferometer (Fig. 3a)[19]. To ensure spectral indistinguishability of the photons we only analysed the light from a narrow emission band around the ZPL (compare Fig. 1a). The measured time-integrated autocorrelation function $\overline{g^{(2)}(\tau)}$ revealed photon bunching for zero time-delay ($\tau \rightarrow 0$) for the fast-decaying ND NV centres, indicating super-Poissonian statistics. We measured values of $\overline{g^{(2)}(0)} > 1$, and as high as $1.14 \pm 0.02$ (Fig. 3b–e). Note that $\overline{g^{(2)}(0)}$ corresponds to the usually quoted $g^{(2)}(0)$. The value of $\overline{g^{(2)}(0)}$ is dependent on the initial state of the system, which in turn is determined by the pre-paration process. Super-Poissonian statistics is not on its own sufficient to indicate cooperative emission, e.g., thermal states also exhibit this. However, the scaling of our autocorrelation measurements is consistent with the assumed initial correlated state in our model (details see Supplementary Note 4). Figure 3e shows the measured value of $\overline{g^{(2)}(0)}$ for the same four representative NDs (ND#1–4) already analysed in Fig. 2, plotted against the corresponding number $N_{max}$ of NV centres acting cooperatively as predicted by our theoretical model; the continuous line shows the upper limit of our theoretical prediction for perfect auto-correlation measurements. In conjunction with the lifetime measurements, these correlation data and the agreement with our theoretical model substantiate the cooperative nature of fluorescence decay in the fastest NDs in our sample.

## Discussion

The observation of SR in a true nanoscale, room-temperature solid-state system paves the way for a wealth of novel research. Immediate subsequent experimental steps include low-temperature studies for obtaining spectral information, and accessing spatial information through, e.g., super-resolution techniques such as stimulated emission depletion (STED) spectroscopy[20]. An obvious extension of our theoretical model incorporates the effect of dipole–dipole interactions which are expected to partially break the cooperativity amongst NV centres[21] but at the same time could allow for super-absorption[10]. In addition, the NDs studied here are a novel system for exploring cooperative atomic forces in the context of optically trapped nanoparticles[12]. Alternative diamond colour centres, such as silicon-vacancy centres[22, 23], exhibit a smaller spread in transition frequencies and much-reduced phononic sidebands—both indicate the potential for greater SR compared with NV centres. Incorporating colour centres in diamond into a microscopic optical cavity[24] could allow the study of the Dicke model in the presence of local processes[25]. Finally, we point out that deterministic implantation techniques[26] and sophisticated material engineering in diamond could enable the controlled creation of mesoscopic superradiant ensembles of colour centres in a given spatial arrangement and with appropriately engineered

photonic[27] and phononic environments[28]. A recent report on superradiant behaviour of two single silicon-vacancy colour centres in a photonic waveguide at helium temperatures constitutes a first step in this direction[29]. Colour centres in diamond might therefore serve as a novel versatile testbed for simulating different regimes of SR over a wide parameter range not easily accessible in other systems.

## Methods

**Nanodiamond sample**. The NDs used in this experiment are synthetic type Ib powders. The ND powder as received (MSY ≤0.1 µm; Microdiamant) was used as the control sample to determine the average baseline value for the lifetime of single NV centres. SR was investigated by using a second diamond powder which had been further treated to increase the concentration of NV centres. The NDs were purified by nitration in concentrated sulphuric and nitric acid (H2SO4-HNO3), rinsed in deionized water, irradiated by a 3-MeV proton beam at a dose of ($1 \times 10^6$ ions per cm$^2$ and annealed in vacuum at 700 °C for 2 h to induce the formation of NV centres (Academia Sinica, Taipei Taiwan[30]). Both the as received and the irradiated NDs were characterised by means of a lab-built confocal scanning fluorescence microscope combined with a commercial atomic force microscope (AFM), described elsewhere[31]. For characterisation, the diamond nanocrystals were dispersed on a 170-µm thick BK7 glass coverslips (BB022022A1; Menzel-Glaser) which had been previously sonicated and rinsed in acetone (C$_3$H$_6$O, purity ≥99.5%; Sigma-Aldrich) for 10 min. The spectral interrogation of the NDs to identify emission from NV centres was performed via a commercial spectrometer (Acton 2500i, Camera Pixis100 model 7515–0001; Princeton Instruments). The size of each individual ND was measured using a commercial atomic force microscope (Ntegra; MT-NDT); the value for the average ND size is ($110 \pm 30$) nm. While for the as received sample the concentration of NV centres is extremely low (at most a few NVs per nanocrystals), for the irradiated one we estimate a concentration of ~ $3 \times 10^6$ NV centres per µm$^3$. This was determined by correlating for nanocrystals of different sizes the average fluorescence intensity measured for each ND with its volume, and cross-checking this ratio with the one given by the sample provider[30].

**Measurements**. Lifetime measurements were performed under off-resonant laser excitation ($\lambda = 532$ nm), for which we employed a pulsed laser source (LDH-P-FA-530; PicoQuant) with the repetition rate set at either 5 or 20 MHz. Emission from the NV centres was filtered either via a long-pass filter (FEL0650, FEL0700; Thorlabs) or via a spectrometer (SpectraPro Monochromator Acton SP2500, dispersion 6.5 nm/mm at 435:8 nm;Princeton Instruments) used as a monochromator and centred around the NVs' ZPL. The emitted photons were detected using an ID Quantique id100–20-ULN single-photon avalanche photodiode detector. In order to achieve higher quantum efficiency, photon-coincidence measurements were performed using a set of two Perkin Elmer SPCM-AQR-14 avalanche photodiode detectors instead. They were arranged in a Hanbury-Brown and Twiss interferometer configuration (Fig. 3a) in order to determine the second-order correlation function $g^{(2)}(\tau)$. For pulsed excitation, we measured the time-integrated autocorrelation function, $\overline{g^{(2)}(\tau)} \equiv \int_{-\tau}^{\tau} dt\langle : I(0)I(t) : \rangle / \int_{-\tau}^{\tau} dt\langle I(0)\rangle\langle I(t)\rangle$, where $\langle I(t)\rangle$ is the luminescence signal intensity. This is evaluated by normalising the photon coincidences of the '0' peak against the other peaks, for time slices of increasing duration. As $\tau \to 0$, $\overline{g^{(2)}(\tau)}$ approaches the standard autocorrelation function $g^{(2)}(0)$, which was measured to identify single NV centres in the control ND sample displaying the characteristic photon anti-bunching dip signifying sub-Poissonian count statistics. On the other hand, superradiant NDs revealed super-Poissonian statistics characterised by photon bunching with a corresponding $\overline{g^{(2)}(0)} \geq 1$ and up to $1.14 \pm 0.02$, (cf. main text, Fig. 3b–d).

**Data availability**. Data available on request from the authors.

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

## Acknowledgements

We thank Marcus Doherty and Neil Manson for helpful discussions. This work was funded by the Australian Research Council Centre of Excellence for Engineered Quantum Systems (CE110001013).

## Author contributions

C.B., M.v.B., M.L.J. and T.V. conceived the experiments. C.B. and M.v.B. performed the experiments. C.B., M.v.B., M.T.J. and B.Q.B. analysed the data. M.T.J., T.V. and G.K.B. constructed the theoretical model, which was further refined by B.Q.B. and R.M. All authors discussed the data and contributed to the writing of the manuscript.

## Additional information

**Competing interests:** The authors declare no competing financial interests.

