## [Peer Review File · Nature Communications]

Reviewers' comments:

Reviewer #1 (Remarks to the Author):

I really enjoyed reading this interesting and clearly written paper. I think it will be well received by the Nature Communications audience.

There are just two questions I have to the Authors:

1. Some years ago, there was a series of very nice papers by James Rabeau and colleagues from the same University, where the effect of the nanodiamond size on NV spontaneous-emission rates was strikingly demonstrated and extensively studied. I think this is directly relevant for this work, but I did not see this discussed. Why ?

2. On Fig. 3d in the main text, the error bars shown are much larger than the spread of the point. I often see this in papers when the authors show the standard deviation instead of the standard error (which is not a good idea). May this be the case here?

Reviewer #2 (Remarks to the Author):

Referee report, manuscript NCOMMS-17-03759-T:

"Room-temperature spontaneous superradiance from single diamond nanocrystals"

In this work the authors explore the possibility to see superradiance in NV centers at room temperature. By considering a large number of samples they find an evident increased emission rate for some samples which is attributed to the collective emission. The experimental results are backed-up by numerical simulations of a Lindblad master-type model. The agreement between experiments and simulations is striking.

The paper is well written and easy to follow. The supplementary material provides a thorough analysis of the underlying mechanisms, which provides a necessary complement for the experimental results.

While the system is extremely complicated, I find their results convincing. The numerical simulations together with the correlation measurements clearly indicate superradiance. Superradiance as such might be a bit dated today, but it is interesting to explore it in NV centers which is a fairly new field. On top, they see the effect at room temperature. I therefore recommend publication.

I have only a few comments to the otherwise very well written manuscript:

1) It is not clear how the present work differs qualitatively from that of Ref. 4. Both are at room temperature, but is it correct that Ref. 4 is not spatially localized?

2) Regarding the local dephasing. This has recently been studied in terms of the Dicke model (Phys. Rev. A 94, 061802 (2016)). This might be a relevant reference.

3) The authors argue why coherence and "collectiveness" can survive at room temperature. Naively one would not guess that it survives at high temperatures. I guess that a more thorough analysis of the theoretical model could show this. Having the full state $\rho(t)$ allows for quantifying coherences and entanglement. Would this be possible? Note that the importance of entanglement for superradiance has been discussed in the past, see for example Phys. Rev. A 84, 023805 (2011).

4) The abbreviation ND has not been defined.

5) In the caption of Fig. 1, the ZPL is said to be around 639 nm. In the figure, however, I interpret the ZPL to be 637 nm.

6) The prospects to demonstrate the Dicke PT in the present model interacting with an optical mode is mentioned. I don't see however how the No-Go theorem would be circumvented here.

Reviewer #3 (Remarks to the Author):

The manuscript by Bradac et al. describes photoluminescence experiments on superradiance from NV-centers in diamond nanocrystals. The central claim is the observation of superradiance and that it occurs at room temperature. The topic is timely and interesting and while the quality of the experimental data is generally high and the underlying theoretical model rather elaborate, the experiment has a number of short-comings, which makes it less than convincing.

The central point is the claimed observation of superradiance. There are a number of problems with this claim:

1. The shape of the decay curve for Dicke superradiance is not expected to be a simple exponential as used in the author's analysis. Since the coherence builds up over time, the decay would be a more complex function. The experimental data therefore does not seem to support the superradiant interpretation. In fact the good single exponential fit seems to directly disprove superradiance.
2. The use of a single exponential fit and associating it with the radiative decay rate is also problematic for more general reasons because there is no way to separate radiative and non-radiative decay processes. Without a microscopic model of the decay dynamics it is not clear what the fitted decay rate means. It is sound that the authors check for other fitting models but that does not resolve this.
3. An important argument is the relation between diamond size and brightness and decay rate (Figs. 2 and 3 in the SI). In Fig. 2, the absence of data in the shaded area is used to argue for the validity of their model. Given the scatter of the data points, this is not extremely convincing but there seems to be a trend that the smallest crystals (red points) behave in different ways than the bigger ones. A number of effects could explain this and I wonder if the authors can rule out the following: Modified far-field pattern due to the different crystal size (would affect Fig. 3), modified photonic environment (would affect both Fig. 2 and 3), and surface effects?
4. The comments above point to another concern: the authors have meticulously measured on 100 nanocrystals but only very few of them display the fast decay rates attributed to superradiance. The central measurements are therefore carried on outliers in the data set. This approach might be justifiable but it is not clear what justifies it here.

In conclusion, while this is clearly in several ways a high-quality work, the shortcomings mentioned above would need to be clarified for the manuscript to be acceptable for publication.

Response to Reviewers

We thank all three referees for carefully reading the manuscript and providing useful comments on our work. All three referees seem to appreciate the content of the paper. Referees 1 and 2 recommend publication and praise the content, clarity and style of writing of the manuscript. Referee 3, however, has some doubts about the validity of our findings, but is in principle open to the idea of the manuscript being published in Nature Communications. In the following, we are going to address all open questions/criticism in detail. We are confident that our answers/corrections will clarify the remaining doubts and hope that the revised manuscript will be considered favourably by referees and by editors. Note that changes made in the text are indicated in blue colour.

Reviewer #1 (Remarks to the Author):

I really enjoyed reading this interesting and clearly written paper. I think it will be well received by the Nature Communications audience.

There are just two questions I have to the Authors:

1. Some years ago, there was a series of very nice papers by James Rabeau and colleagues from the same University, where the effect of the nanodiamond size on NV spontaneous-emission rates was strikingly demonstrated and extensively studied. I think this is directly relevant for this work, but I did not see this discussed. Why?

The papers discussed by Rabeau and colleagues explore the role of the nanodiamond host on the decay rate of spontaneously emitting NV centres. One of the key papers here is APPLIED PHYSICS LETTERS 102, 253109 (2013), which investigates the emission dynamics of single NV centres as the size of their ND host is reduced to 50nm and below (starting from around 250nm).

There are two key findings in there: 1) compared to the bulk rate the emission rate of the NV decreases as the ND size is reduced, until 2) the NV centre gets too close to the surface such that non-radiative processes kick in and the rate shoots up again. The key difference of our work to the above paper (and the reason for not citing it) is the high density of NV centres within a single ND. Hence our fluorescence signal is dominated by “bulk-like” NV centres away from the surface such that surface effects do not play a significant role. This is consistent with the fact that in our sample small NDs (down to 50nm size) do not show a significant speed-up. What we do observe, however, is the fact that for most NDs (the ones not showing significant speed-up due to superradiance), the emission rate is greatly reduced compared to true bulk systems. We attribute that to the reduced optical density of states in a ND, which is also the explanation for point 1) above. This is well known and has been observed in many instances.

2. On Fig. 3d in the main text, the error bars shown are much larger than the spread of the point. I often see this in papers when the authors show the standard deviation instead of the standard error (which is not a good idea). May this be the case here?

We thank the referee for pointing out this important issue. We agree that the relevant quantity is the standard error of the mean rather than the standard deviation of the underlying distribution. We have corrected both Figures 3d and 3e accordingly. Note that the error bars are non-uniform across the data points in d) and e) since the sample sizes were not the same.

Reviewer #2 (Remarks to the Author):

Referee report, manuscript NCOMMS-17-03759-T:

"Room-temperature spontaneous superradiance from single diamond nanocrystals"

In this work the authors explore the possibility to see superradiance in NV centers at room temperature. By considering a large number of samples they find an evident increased emission rate for some samples which is attributed the collective emission. The experimental results are backed-up by numerical simulations of a Lindblad master-type model. The agreement between experiments and simulations is striking.

The paper is well written and easy to follow. The supplementary material provides a thorough analysis of the underlying mechanisms, which provides a necessary complement for the experimental results.

While the system is extremely complicated, I find their results convincing. The numerical simulations together with the correlation measurements clearly indicates superradiance. Superradiance as such might be a bit dated today, but it is interesting to explore it in NV centers which is a fairly new field. On top, they see the effect at room temperature. I therefore recommend publication.

I have only a few comments to the otherwise very well written manuscript:

1) It is not clear how the present work differ qualitatively from that of Ref. 4. Both are at room temperature, but is it correct that Ref. 4 is not spatially localized?

Reference 4 reports the observation of superradiance from a room-temperature molecular gas. Our experiment differs in several critical ways. i) Most notably, the NV centres are localized defects in a solid-state crystal. ii) The density of NV centres in our nanodiamonds ($\sim 10^{18}$ emitters/cm³) vastly exceeds that achievable in atomic gases ($\sim 10^{13}$ atoms/cm³). iii) Due to this incredibly high density, we meet the condition in Dicke's original proposal that many emitters are localized to within a single cubic wavelength.

2) Regarding the local dephasing. This has recently been studied in terms of the Dicke model (Phys. Rev. A 94, 061802 (2016)). This might be a relevant reference.

We thank the referee for pointing out a relevant reference, which we have included as a new reference [25] replacing the previous one.

3) The authors argue why coherence and "collectiveness" can survive at room temperature. Naively one would not guess that it survives that high temperatures. I guess that a more thorough analysis of the theoretical model could show this. Having the full state $\rho(t)$ allows for quantifying coherences and entanglement. Would this be possible? Note that the importance of entanglement for superradiance has been discussed in the past, see for example Phys. Rev. A 84, 023805 1 (2011).

This is an interesting possibility. The Wiegner, von Zanthier, and Agarwal paper shows that a collection of spins prepared in an entangled state of shared excitations, will provide a characteristic signature in the directionality of the emitted radiation intensity due to the interference of the pathways from each spin to a detector. The angular width of the interference term in the intensity scales like the inverse of the number of spins times their average separation divided by the wavelength. There they assume an ordered sample where the spins are arranged on a chain but it would be worth investigating whether the NV centres in our samples are ordered enough to make this phenomenon observable using detectors with fine angular resolution. Another possibility is to transfer entanglement with respect to optical excitation to entanglement in the ground spin-1 electronic states of the NV centres that are addressable with microwave fields. The existence of decay channels that are non-spin conserving may allow for transferring a state with shared optical excitation along one spin polarization direction (e.g. $|W\rangle$ states with spin projection $m=0$ in optical ground and excited) to entangled

states which are $|W\rangle$ states but with shared $m=0$ and $m=1$ ground states. This would require that the decay process gains little information about the location of the spin in order to preserve the permutation symmetry. While these questions are highly interesting and certainly subject of future studies, they go well beyond the scope of the present paper. We therefore prefer not to include them in the present paper.

4) The abbreviation ND has not been defined.

The abbreviation ND has now been defined in the first, introductory paragraph.

5) In the caption of Fig. 1, the ZPL is said to be around 639 nm. In the figure, however, I interpret the ZPL to be 637 nm.

We appreciate the referee's attention to detail. The ZPL in the literature for bulk diamond is indeed 637nm, but it tends to be shifted in nanodiamonds due to the different nanoscale environment and the associated crystal stress. Across our samples, the ZPL was consistently around 639nm, which we use as the representative wavelength in this Figure caption. We have adjusted the caption in Figure 1 to make this clearer.

6) The prospects to demonstrate the Dicke PT in the present model interacting with an optical mode is mentioned. I don't see however how the No-Go theorem would be circumvented here.

Based on the referee's comment, we have revisited the issue and agree that our system does not circumvent the no-go theorem. The text has been modified to:

"Incorporating colour centres in diamond into a microscopic optical cavity [24] could allow the study of the Dicke model in the presence of local processes [25]."

Note: The previous reference [25] was replaced with the reference mentioned by the referee in point 2) above.

Reviewer #3 (Remarks to the Author):

The manuscript by Bradac et al. describes photoluminescence experiments on superradiance from NV-centers in diamond nanocrystals. The central claim is the observation of superradiance and that it occurs at room temperature. The topic is timely and interesting and while the quality of the experimental data is generally high and the underlying theoretical model rather elaborate, the experiment has a number of short-comings, which makes it less than convincing.

The central point is the claimed observation of superradiance. There are a number of problems with this claim:

1. The shape of the decay curve for Dicke superradiance is not expected to be a simple exponential as used in the author's analysis. Since the coherence builds up over time, the decay would be a more complex function. The experimental data therefore does not seem to support the superradiant interpretation. In fact the good single exponential fit seems to directly disprove superradiance.

We did not assume the decay curve was an exponential nor did exponential fit the measured decay curves. Our model involved a full superradiance analysis with collective effects and additional dephasing, as detailed in the Supplementary. The resulting rate equations (see Eq 11 in the Supplementary) produce decay curves that fit the experimental data and show exponential decay only when the number of collective emitter is $N=1$. We also ruled out two alternative models/fit curves, stating in the main text: "In order to test quantitatively against other possible explanations for the observed fast decay dynamics, we also attempted to fit the observed fluorescence decay curves with both a bi-exponential and a deformed exponential [18] decay, both of which gave clearly worse results."

In order to sort the nanodiamonds into classes according to decay speed, we took the peak of the lifetime decay curve, then the value three nanoseconds later, then fit an exponential decay to these two points. This does not produce an exponential fit that matches the entire curve, but it does provide a standardised measure to compare diamonds by speed.

We have modified and added the following text to the main manuscript to clarify our procedure:

“The red curve shows the exponential decay for a single NV centre used for reference. The faster diamonds were not well fit by exponentials, but rather require a superradiant model as described below. However, to make speed comparisons we extracted indicative $1/e$ -lifetimes by fitting a standard exponential to the first three nanoseconds of each decay curve. For the fastest diamonds we observed lifetimes around 1 ns or even below, never reported before for NV centres (e.g. Fig. 1b, ND#4).”

2. The use of a single exponential fit and associating it with the radiative decay rate is also problematic for more general reasons because there is no way to separate radiative and non-radiative decay processes. Without a microscopic model of the decay dynamics it is not clear what the fitted decay rate means. It is sound that the authors check for other fitting models but that does not resolve this.

As described in our response above, a single exponential decay was not assumed. Our model consisted of an extensive set of rate equations, derived from a microscopic model that includes collective effects, local dephasing, and dark decay. For the dark and radiative decay channels we used the best available values from the literature (Figure 1 in the Supplementary shows the specific decay channels we used in our model, both radiative and dark).

The fact that our model correctly fitted all decay curves with collective domain size as the dominant free parameter while holding the dark decay rates fixed according to the literature values suggests that unknown dark decay processes do not seem to play a major role in our experiment.

Further, we did investigate the hypothesis that (dark) decay rate speedups can arise from crystal damage. Section III (and Figure 2) in the Supplemental discusses this hypothesis and rules it out.

3. An important argument is the relation between diamond size and brightness and decay rate (Figs. 2 and 3 in the SI). In Fig. 2, the absence of data in the shaded area is used to argue for the validity of their model. Given the scatter of the data points, this is not extremely convincing but there seems to be a trend that the smallest crystals (red points) behave in different ways than the bigger ones. A number of effects could explain this and I wonder if the authors can rule out the following: Modified far-field pattern due to the different crystal size (would affect Fig. 3), modified photonic environment (would affect both Fig. 2 and 3), and surface effects?

It is important to note that the far-field radiation pattern for different nanodiamonds is irrelevant for the lifetime/decay curves. Rather, it only enters as a factor in those plots relating to relative brightness. These data are used only as a “cross-reference” to the lifetime data, and are not used to draw independent conclusions.

The photonic environment certainly plays a role, and is allowed for in our model. We assume the local density of states is affected by size which will alter the free-field optical decay rate, and so we allow a scaling factor for this optical decay rate. We are sure this plays a role since the optical decay rate we measured was longer in the nanodiamonds than in bulk diamond in almost all cases. This is what one would expect from restricting possible allowed modes by making the diamonds smaller.

We did not notice a trend where smaller NDs had slower base (i.e. single NV centre, non-collective) radiative decay rates than larger NDs, however – nanodiamonds in general are slower than bulk, but within our ensemble size this made no statistical difference to inferred optical rates. Rather, we attribute the fact that smaller NDs were not fast to surface effects resulting in breaking indistinguishability and reducing domain

size. This makes sense – in smaller NDs a greater proportion of NV centres are closer to the surface and will experience random interactions that break indistinguishability.

4. The comments above point to another concern: the authors have meticulously measured on 100 nanocrystals but only very few of them display the fast decay rates attributed to superradiance. The central measurements are therefore carried on outliers in the data set. This approach might be justifiable but it is not clear what justifies it here.

We absolutely agree that the nanodiamonds we studied exhibited a range of cooperativity and we have chosen to focus on the behaviour of a few with reproducible fast decay rates to highlight the superradiance phenomenon. We expect that the reason for this variety is inhomogeneities in the samples in terms of local crystal fields and density and distribution and orientation of colour centres which impact the effects of dipole-dipole interactions. We plan future experiments at lower temperature, and using near resonant absorption spectroscopy, in order to isolate the dominant mechanisms that affect the cooperative behaviour. Concurrently, we are developing a more detailed theoretical model.

We have added the following text to the main manuscript to clarify why many diamonds are not expected to be superradiant:

“We note that collective superradiance requires indistinguishability of the emitters, which can be broken in many ways in a solid-state setting. The fact that the majority of NDs in our ensemble did not exhibit superradiant behaviour suggests that environment seen by the NV centres included many inhomogeneous effects. We attribute the majority of the inhomogeneity to the material preparation which involves high-dose proton irradiation followed by annealing (see Methods).”

We would like to stress that our theoretical model involving collective Dicke states was applied to and successfully fit the decay curves for every nanodiamond in our sample, fast and slow. The foundation of the model is that NVs centres act collectively in domains of indistinguishability (spectral, spatial); that is, not all the NVs in a crystal act together collectively. For each nanodiamond this breaks all the NVs up into indistinguishable domains. Our results suggest that it is somewhat unlikely for a typical nanodiamond in our sample to have many domains of size significantly greater than one (i.e. sizeable collectivity).

Our model captures this phenomenon by allowing domain size to vary as the free parameter. When domain sizes are small, the model matches the slow diamonds. With larger and larger domains, the model accurately tracks the faster diamonds, making the transition from simple exponential decay to superradiance. Again, we stress that our model captures this transition and fits the observed lifetimes better than exponential, bi-exponential, or deformed exponential fits, and does so with minimal free parameters.

An additional mechanism that can inhibit superradiance is dipole-dipole interactions between the NV centres. This effect is described by Eqns (18) – (20) in the supplementary. We are currently carrying out further research to quantify this effect, as well as the rate at which interactions between local and global phononic baths destroy indistinguishability.

In conclusion, while this is clearly in several ways a high-quality work, the shortcomings mentioned above would need to be clarified for the manuscript to be acceptable for publication.

We appreciate the time and thoughtfulness of the referee for providing feedback on our manuscript, and the opportunity he/she gave us to clarify some of the misunderstandings in the presentation of our data analysis. We think that these comments have enabled us to improve the clarity of our manuscript. We hope that the revised manuscript is now convincing and acceptable for publication.

REVIEWERS' COMMENTS:

Reviewer #1 (Remarks to the Author):

I have examined the revised manuscript and the Authors' responses to the comment of all three referees. I think all the questions have been adequately answered and I recommend the paper for publication in its current form.

Reviewer #2 (Remarks to the Author):

Second report, manuscript NCOMMS-17-03759A:

"Room-temperature spontaneous superradiance from single diamond nanocrystals"

After reading the response and revised version I support publication of the manuscript in its present form. Like the third referee, I was at first also skeptical that the measured results demonstrate superradiance. However, the numerical simulation is convincing and the problem at hand is complex and one cannot expect clean and very transparent evidences.

Reviewer #3 (Remarks to the Author):

The authors have carefully revised the manuscript and answered the questions raised. The answers are clear and substantial and I recommend publication. As a minor remark, I suggest expanding the explanations of the theoretically predicted decay curves as well as the fitting procedure in the supplementary information, cf. the authors' response to my question.